# AI-augmented intraoperative decision-making workflows in diffuse midline glioma biopsy using cryosection pathology

Yifan Yuan [1,2,8], Qi Yue [1,8] ✉, Yining Wang[1,8], Zunguo Du[3,8], Xingfu Wang[4,8], Fang Yan [2], Haixia Cheng[3], Kaitao Chen[2], Zanyi Wu[5], Jianping Song[1,5,6], Chunfeng Song[2], Lei Bai[2], Mu Zhou [7] ✉, Mianxin Liu [2] ✉ & Ying Mao [1] ✉

Cryosection pathology is essential for intraoperative diagnosis of diffuse midline gliomas, yet it often leads to diagnostic errors and may prompt unnecessary re-biopsies before completion of the formal molecular assessment. In this study, we propose an AI-augmented framework, CryoAID, for rapid molecular outcome prediction during surgery for patients with diffuse midline glioma. CryoAID integrates a generative model to correct cryosection artefacts and a pathology foundation model to predict molecular statuses directly from cryosection images. We validate CryoAID across multiple cohorts to predict tumoural molecular statuses in the internal ($n = 326$), external multi-centre ($n = 52$), and consecutive ($n = 68$) datasets. In particular, CryoAID accurately predicts major molecular statuses (e.g., ATRX, H3K27M, and TP53) using cryosection images that were previously deemed disqualified for molecular examinations. Beyond tumour cells, CryoAID reveals highly differential clinical features, including glial cell proliferation, abundant cytoplasm, and localised endothelial proliferation. In the retrospective analyses, CryoAID reduces re-biopsy rates by 26.4% and 26.6% in the internal and consecutive datasets, respectively. Our findings demonstrate that the AI-augmented pathology workflow can extract diagnostic value from specimens previously considered non-viable by traditional histopathology. This approach represents a shift towards real-time molecular pathology, potentially reducing re-biopsies and improving diagnostic precision for patients with diffuse midline glioma.

Diffuse midline gliomas (DMGs) predominantly affect children and adolescents and have become the leading cause of death among paediatric brain tumours[1-3]. Major molecular biomarkers, including H3K27M[4], TP53[5], ATRX[6], and BRAF[7], serve as independent prognostic factors and therapeutic targets for improving patient survival[8,9]. In the surgical setting, a timely assessment of these biomarkers remains difficult in the surgical environment. The deeply infiltrative nature of DMGs, combined with their location near vital brain regions, makes the

---

[1]Department of Neurosurgery, National Centre for Neurological Disorders, Huashan Hospital, Fudan University, Shanghai, China. [2]Shanghai Artificial Intelligence Laboratory, Shanghai, China. [3]Department of Pathology, Huashan Hospital, Fudan University, Shanghai, China. [4]Department of Pathology, National Regional Medical Centre, The First Affiliated Hospital Binhai Campus, Fujian Medical University, Fuzhou, Fujian, China. [5]Department of Neurosurgery, National Regional Medical Centre, The First Affiliated Hospital Binhai Campus, Fujian Medical University, Fuzhou, China. [6]Department of Neurosurgery, National Regional Medical Centre, Huashan Hospital Fujian Campus, Fudan University, Fuzhou, Fujian, China. [7]Department of Computer Science, Rutgers University, New Brunswick, New Jersey, USA. [8]These authors contributed equally: Yifan Yuan, Qi Yue, Yining Wang, Zunguo Du, Xingfu Wang. ✉e-mail: yueqi1989@126.com; muzhou1@gmail.com; liumianxin@pjlab.org.cn; maoying@fudan.edu.cn

surgical removal highly challenging and hazardous. Therefore, clinical practice routinely combines biopsy and intraoperative frozen section examination with comprehensive molecular analysis for diagnostic confirmation[10,11].

The neurosurgical biopsy process involves a complex interaction between the pathologist and the surgeon, constrained by limited operative time, tissue availability, and imaging quality. In this process, only qualified biopsy samples are selected for detailed molecular diagnoses using immunohistochemistry (IHC) and sequencing[12–14]. During surgery, pathologists must make timely decisions on the biopsy samples, typically classifying them as either *Pass* or *No-Pass* samples. This assessment directly influences the surgeon's decision to continue or terminate the surgical operation. However, the accuracy of such real-time evaluations is often compromised by the inherent challenge in cryosection pathology. Two common scenarios frequently result in undesired No-Pass outcomes, thereby increasing the risk of re-biopsy: (i) artefacts or atypical cellular morphology introduced during cryosectioning may obscure tumour cells, preventing their identification; and (ii) even when tumour tissue is present, suboptimal visual cues from cryosections can reduce diagnostic confidence and prompt surgeons to request additional samples.

Reducing unnecessary re-biopsies is particularly critical for DMGs as multiple biopsies significantly heighten the risk of patient disability or death[15]. Two major challenges persist in the current frozen section practice. First, the precise molecular profiling has not been fully extended to the intraoperative frozen section evaluation. Formalin-fixed, paraffin-embedded (FFPE) specimens remain the primary source for immunohistochemical (IHC) and molecular analyses[16–18]. However, the FFPE procedure is time-consuming and precludes rapid feedback via the trial-and-error operation. Second, it remains unclear which histological features from cryosectioning can reliably inform molecular diagnostics. The current "Pass" criterion primarily relies on visual detection of sufficient tumour cells, without accounting for the link between histopathological appearance and molecular profiles. Emerging studies indicate that genetic signals of tumour can be detected in adjacent non-tumour tissues[19], indicating that intraoperative decision-making for cryosectioned samples requires a more nuanced assessment of tissue-level imaging features.

Clinical artificial intelligence (AI) has advanced the characterization of cryosection images and accelerated molecular diagnosis[20,21]. The use of generative AI promises to enhance the assessment of cryosection pathology and mitigate cryosection-induced artifacts[22]. For instance, pre-trained foundation models excel at extracting subtle image features that are challenging to discern with the naked eye, thereby augmenting the capacity of molecular diagnostics in clinical examinations[23–26]. Despite their potential clinical impact, these advances have primarily focused on FFPE tissue analysis[27–29]. Extending AI-based prediction directly to frozen sections would enable pathologists to receive immediate diagnostic support, facilitating sample adequacy assessment, tissue abnormality detection, and reducing unnecessary re-biopsies.

In this study, we explore the clinical impact of AI-driven assessment for DMG patients by proposing the CryoAID as a Cryosection-based AI for Intraoperative Diagnosis (Fig. 1A, B). Our primary objective is to enable AI-powered rapid molecular diagnostics with interpretable visual cues, both on the "Pass" cryosectioned pathology samples and on difficult samples previously deemed "No-Pass" by clinical experts. CryoAID aims to repurpose No-Pass samples and reduce unnecessary re-biopsies within the routine diagnostic process of cryosectioned pathology. We evaluate a retrospective multi-centre collection of cryosection pathology data from the DMG biopsy operation. We illustrate that using CryoAID could potentially reduce the re-biopsy triggered by unnecessary "No-Pass" decisions, thereby improving the intraoperative decision-making and minimising the surgical risks of patients with DMG.

## Results

### CryoAID strongly predicts mutant statuses in Pass images

This experiment involved passing cryosection images to demonstrate the capacity of CryoAID for predicting major mutant statuses (ATRX, H3K27M, and TP53. Fig. 1D Part I). In Fig. 2A–C, we see that CryoAID demonstrates strong improvements over all other methods with statistical significance. For the tasks of predicting TP53, H3K27M, and ATRX, CryoAID obtains area under the curve (AUC) = $0.866 \pm 0.051$ (95% confidence interval, CIs = [0.828, 0.905]), AUC = $0.788 \pm 0.044$ (CIs = [0.754, 0.821]), and AUC = $0.774 \pm 0.048$ (CIs = [0.737, 0.810]). These AUCs from CryoAID are all significant for the three genes (Fig. 2D), along with positive predictions in terms of multiple performance metrics (Fig. 2G). On ARTX, CryoAID exhibits accuracy (ACC) = $0.709 \pm 0.054$ (CIs = [0.668, 0.750]), F1 = $0.654 \pm 0.052$ (CIs = [0.614, 0.693]), sensitivity (SEN) = $0.656 \pm 0.078$ (CIs = [0.597, 0.714]), and specificity (SPE) = $0.730 \pm 0.070$ (CIs = [0.678, 0.783]). For H3K27M, CryoAID has ACC = $0.742 \pm 0.040$ (CIs = [0.712, 0.773]), F1 = $0.789 \pm 0.024$ (CIs = [0.771, 0.808]), SEN = $0.708 \pm 0.040$ (CIs = [0.678, 0.739]), and SPE = $0.780 \pm 0.052$ (CIs = [0.741, 0.819]). For TP53, the metrics are ACC = $0.800 \pm 0.054$ (CIs = [0.759, 0.841]), F1 = $0.798 \pm 0.053$ (CIs = [0.759, 0.838]), SEN = $0.762 \pm 0.064$ (CIs = [0.713, 0.810]), and SPE = $0.826 \pm 0.058$ (CIs = [0.783, 0.870]).

We evaluated the generalizability of CryoAID using the consecutive testing dataset and the multi-centre external testing dataset. In Fig. 2E, under consecutive external testing, CryoAID accurately detects the TP53, H3K27M, and ATRX statuses with AUC = 0.772 (CIs = [0.656, 0.887], $p < 0.001$), AUC = 0.707 (CIs = [0.581, 0.834], $p = 0.001$), and AUC = 0.662 (CIs = [0.529, 0.795], $p = 0.017$). Correspondingly, the other performance metrics are ACC = 0.716, F1 = 0.689, SEN = 0.700, SPE = 0.730, $p < 0.001$ for H3K27M, and ACC = 0.716, F1 = 0.740, SEN = 0.677, SPE = 0.750, $p < 0.001$ for TP53 (Fig. 2H). The metrics for ATRX are relatively restricted but significant (ACC = 0.597, SEN = 0.560, SPE = 0.619, $p < 0.05$; F1 = 0.562, $p < 0.001$). And in the multi-centre testing (Fig. 2F, G), on the combined result (centre-wise results in Fig. S2), the model's prediction capacity remains positive in ATRX (AUC = 0.789, CIs = [0.665, 0.914], $p < 0.001$), H3K27M (AUC = 0.793, CIs = [0.668, 0.919], $p < 0.001$), and TP53 (AUC = 0.840, CIs = [0.728, 0.951], $p < 0.001$). In the challenging multi-centre evaluation (Fig. 2I), the performance metrics remain favourable, where CryoAID yielded ACC = 0.765, F1 = 0.647, SEN = 0.769, SPE = 0.763, $p < 0.001$ for ATRX, ACC = 0.725, F1 = 0.727, SEN = 0.696, SPE = 0.750, $p < 0.001$ for H3K27M, and ACC = 0.765, F1 = 0.776, SEN = 0.750, SPE = 0.778, $p < 0.001$ for TP53. Overall, CryoAID, based on Pass cryosection images, displays a high capability in the task of molecular diagnoses under various external settings.

In Fig. 3, we visualized the predictive features for the H3K27M mutant from CryoAID. From Fig. 3A, B, scant tissue displays diffusely infiltrating tumour cells with prominent cytoplasmic processes and small round nuclei, exhibiting nuclear atypia, high cellular density, and rare mitotic figures. In the 2021 WHO Classification[30], the H3K27M-mutant DMGs are all assigned grade 4 gliomas with astrocytic morphology in histopathology. This consistent evidence supports that CryoAID enables clinically relevant feature detection in cryosection images of the H3K27M mutant.

### CryoAID enables the molecular diagnosis in No-Pass images

We explored the challenging setting by using CryoAID to predict mutant status on No-Pass images in two different scenarios (Fig. 1D Part II). First, we directly transferred models trained on Pass images to evaluate on the No-Pass images (Fig. 4). Second, we re-trained another set of models based solely on No-Pass images (Fig. 5). We assessed the significance of the predictability and investigated the predictive features in No-Pass images in Fig. 5.

In Fig. 4A-C, we observe that our model can detect mutant statuses of ATRX, H3K27M, and TP53 in No-Pass images using

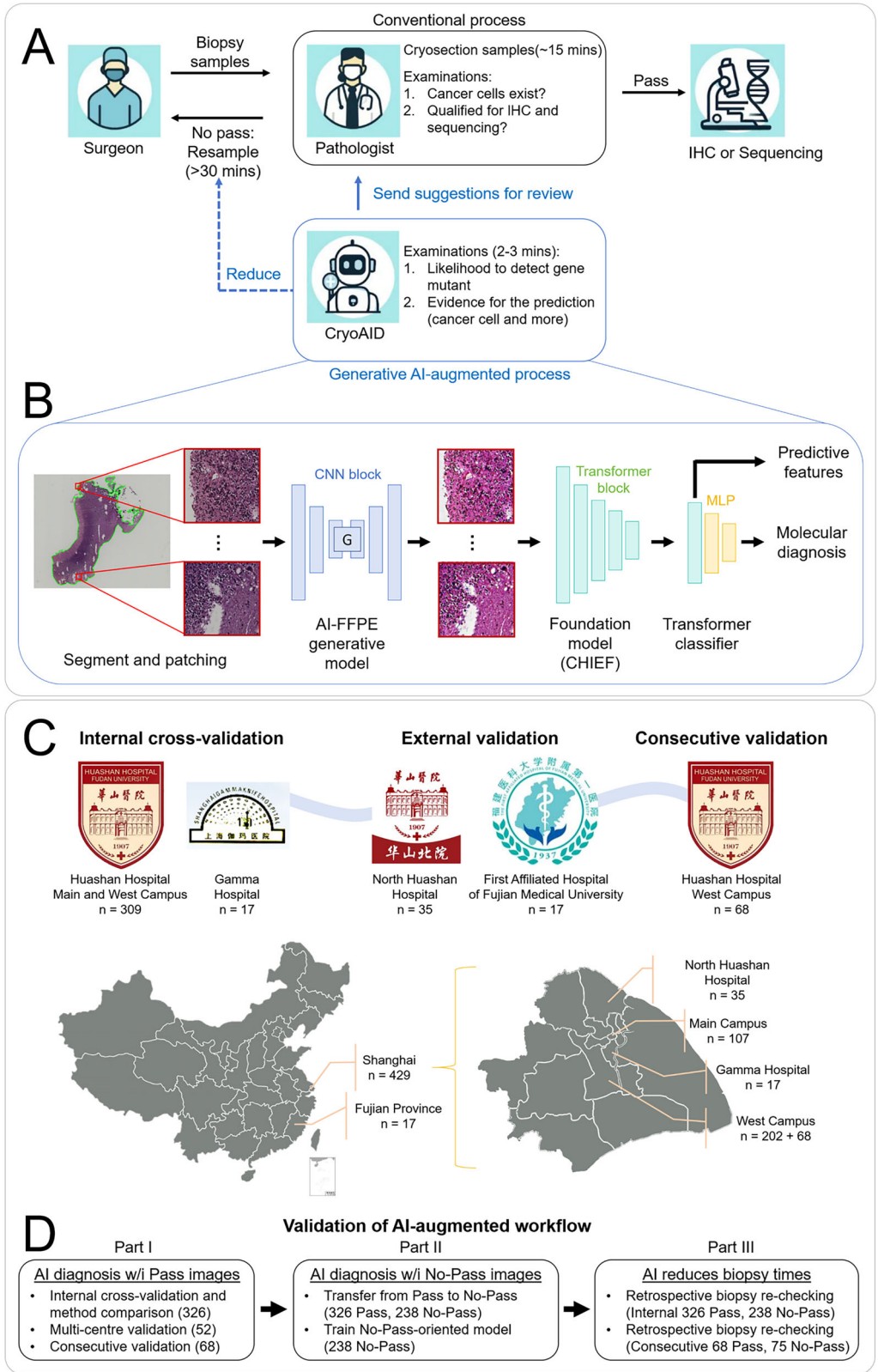

Pass-image pre-trained models. Although AUC values are relatively low in this challenging setting, the significance levels are statistically satisfactory (AUC = 0.557, CIs = [0.479, 0.634], $p = 0.071$ for ATRX; AUC = 0.728, CIs = [0.665, 0.790], $p < 0.001$ for H3K27M; AUC = 0.732, CIs = [0.665, 0.798], $p < 0.001$ for TP53). This observation suggests that certain predictive features are shared in Pass and No-Pass images.

We hypothesized that these shared features could be the tumour cells that escaped the visual examinations. To evaluate, we selected the cryosectioned slides with the H3K27M mutant being accurately detected and sent the corresponding predictive patches to the pathologist for visual re-checking. The pathologist identified malignant, uncertain, and benign patches, which were exemplified in Fig. 4D. The statistics support that most of the predictive patches are

**Fig. 1 | CryoAID enhances the efficiency of the clinical biopsy workflow and reduces the surgical risk of DMG patients. A** In the conventional biopsy process, after receiving the specimen from the surgeon, the pathologist decides whether the cryosection slide is "Pass" or "No-Pass." Due to the limited image quality of cryosection slides and the lack of clear visual features to ensure molecular adequacy, unnecessary "No-Pass" decisions are frequently made. This often leads to significant time delays (exceeding 30 minutes) and increases biopsy-related risks for patients with DMG. The proposed CryoAID provides both the likelihood of molecular diagnosis for a given slide and corresponding visual evidence. CryoAID's output can trigger re-examination of slides, allowing pathologists to revise incorrect "No-Pass" decisions to "Pass," thereby potentially reducing unnecessary re-biopsies. **B** The algorithmic workflow of CryoAID. CryoAID employs an AI-FFPE generative model to enhance the visual quality of cryosection slides and leverages a pathology-specific pre-trained foundation model to extract reliable

representations. The slide-level features are integrated via a Transformer-based classifier to generate probabilistic predictions of gene mutation status. **C** Multi-centre data collection for internal cross-validation, external validation, and consecutive validation datasets. **D** The detailed validation pipeline. The three components correspond to the major result sections (I–III). Part I follows the standard clinical workflow, focusing on AI-based molecular diagnosis using "Pass" cryosection images. Part II demonstrates CryoAID's ability to repurpose "No-Pass" images to improve molecular diagnostic performance. This part includes two experiments: transfer learning and direct training. Part III jointly uses both "No-Pass" and "Pass" images to evaluate CryoAID's capability to reduce re-biopsies through retrospective re-checking analysis. CryoAID re-examines images in the same sequence as in the surgical workflow and provides predictive assessments with reduced biopsy counts based on combined "Pass" and "No-Pass" inputs.

---

malignant (Fig. 4E, 715/1035, 69.1%). Malignant patches demonstrate moderately dense infiltrates of oval tumour cells with marked nuclear atypia and prominent cytoplasmic processes. Mitotic figures are readily identifiable, with focal vascular endothelial proliferation observed in some cases. Most of the tumour cells in these re-identified malignant patches appear in the tissue boundary, where the tumour cells can be hidden from previous examinations of pathologists. The uncertain patches manifest broken tissue showing increased cell density, formation of foamy cells in sheets accompanied by glial proliferation, a few small round cells with projections scattered throughout, and some nuclei being slightly atypical. The benign patches involve normal cell morphology or slight swelling of some cell cytoplasm, with no obvious nuclear atypia. As we expected, the tumour-related features can be detected in a large part of cryosectioned slides (Fig. 4F, 60/69, 86.9%), potentially caused by the missed examination. However, there also exists a certain amount of predictive patches (Fig. 4E, uncertain: 212/1035, 20.5%; benign: 108/1035, 10.4%) and slides (Fig. 4F, 3.1%) being not tumour-related.

In Fig. 5A–C, we display that in the No-Pass-based training and predictions, AUCs range from 0.736 to 0.793 and *P*-values are all less than 0.001. Note that the AUC for ATRX is higher than that in Fig. 5A, supporting that without learning on Pass images, CryoAID can identify a set of specific features associated with the mutant status. In other words, there exist predictive features specific to No-Pass images that are not strongly related to tumour tissues.

We explored No-Pass image-specific features using a representative case of a 19-year-old female patient with a glioma located in the bilateral pons and medulla oblongata and confirmed H3K27M mutant. Due to the tumour's diffuse infiltration within the brainstem, the pontine lesion was targeted for biopsy (Fig. 5D). Previous intraoperative cryosection pathology analysis indicated glial proliferation with mild nuclear atypia, recommending further sampling (No-Pass image shown in Fig. 5D). Following the analysis of CryoAID with No-Pass-based training, the tissue imaging exhibited indications of H3K27M mutation. In Fig. 5E, we observe a small proportion of nuclear atypia cells (i.e., suspected tumour cells), a rich presence of glial cells, with localised evidence of endothelial cell proliferation (Fig. 5F, G). In the meantime, we confirmed that this H3K27M mutation was identified via IHC imaging from the same sample (Fig. 5H). While CryoAID successfully detected suspected tumour cells in this low-density case, traditional pathology would consider such sparse tumour cells insufficient for confident diagnosis. Notably, CryoAID overcame this limitation by extracting information largely from the tumour's surrounding tissues—including the abundant glial cells and endothelial proliferation—rather than relying solely on the scarce tumour cells themselves, thereby achieving accurate mutant status prediction despite the challenging sample.

### A retrospective biopsy re-checking with assistance of CryoAID
CryoAID could generate rapid decision-making support during the biopsy operation on both No-Pass and Pass images. In this analysis, we

retrained models based on all Pass and No-Pass samples. The models were applied to a retrospective biopsy re-checking analysis, mimicking intraoperative conditions (Fig. 1D Part III. Illustration for details in Fig. 6A) under a simplified setting. In particular, we strictly maintained the arriving order of the collected retrospective cryosection images and sequentially input them to our established models. Then CryoAID independently assessed the mutant status of each arriving sample. We compared the actual biopsy counts by humans and AI detection counts for any mutant correctly, as mutations in any of the three diagnostic genes (ATRX, H3K27M, and TP53) suggest tumour presence. Please note that this statistical process may be optimistic to a certain extent, as in real-world practice, there is no ground-truth mutation status available during surgery, only the pathologist's judgment. We expect that the actual biopsy counts achieved through AI-pathologist collaboration will exceed the estimation in this analysis, but remain lower than those from human experts alone.

In Fig. 6B, we observe a range of significant predictability from such trained models. Using the information about sampling order, we were able to calculate the prediction performance of CryoAID within the first two times of biopsies (Fig. 6D, E and Table S2). By properly adjusting a decision threshold, using once sampling ($n = 315$), the model can obtain 0.675–0.733 ACC and 0.722–0.781 AUC for different mutant predictions. Under twice sampling ($n = 173$), the model yields further increased performances, with 0.680-0.775 ACC and 0.777–0.844 AUC. We see that CryoAID underpins the intraoperative decision support and reduces re-biopsy. In Fig. 6F, in the internal dataset, the averaged biopsy counts for humans are 1.73, while expected to be 1.27 for "AI". The AI process would significantly reduce 149 times of biopsy out of 564 times in total (326 patients, the reduction ratio is 26.4%, $p = 2.89e-26$, t(325) = 11.52, one-sided paired t-test).

We further re-trained a set of models based on all internal data and applied them to all consecutive testing data. The predictive performance for the three genes remains satisfactory (Fig. 6C, ATRX: AUC = 0.604, CIs = [0.508, 0.700], $p = 0.017$; H3K27M: AUC = 0.704, CIs = [0.619, 0.789], $p < 0.001$; TP53: AUC = 0.753, CIs = [0.672, 0.834], $p < 0.001$)). In Fig. 6G, using the consecutive data, the averaged biopsy counts for "human" are 2.10 and 1.54 for "AI". The AI process could significantly save 38 times of biopsy out of 143 times in the 68 patients (reduces 26.6%, $p = 6.61e-10$, t(67) = 7.03). Note that the patients with 6 and 4 times of biopsy in the internal and consecutive datasets are confirmed with no mutant, and thus AI can not reduce the re-biopsy. Despite the model advance, this indicates the existence of extreme cases that cannot benefit from CryoAID.

### Dual validation for low-grade midline gliomas
It is noteworthy that a subset of midline gliomas, including pilocytic astrocytomas (PAs), are low-grade, underscoring the substantial differences in clinical management strategies. PAs require maximal resection, whereas the presence of an H3K27M mutation could make

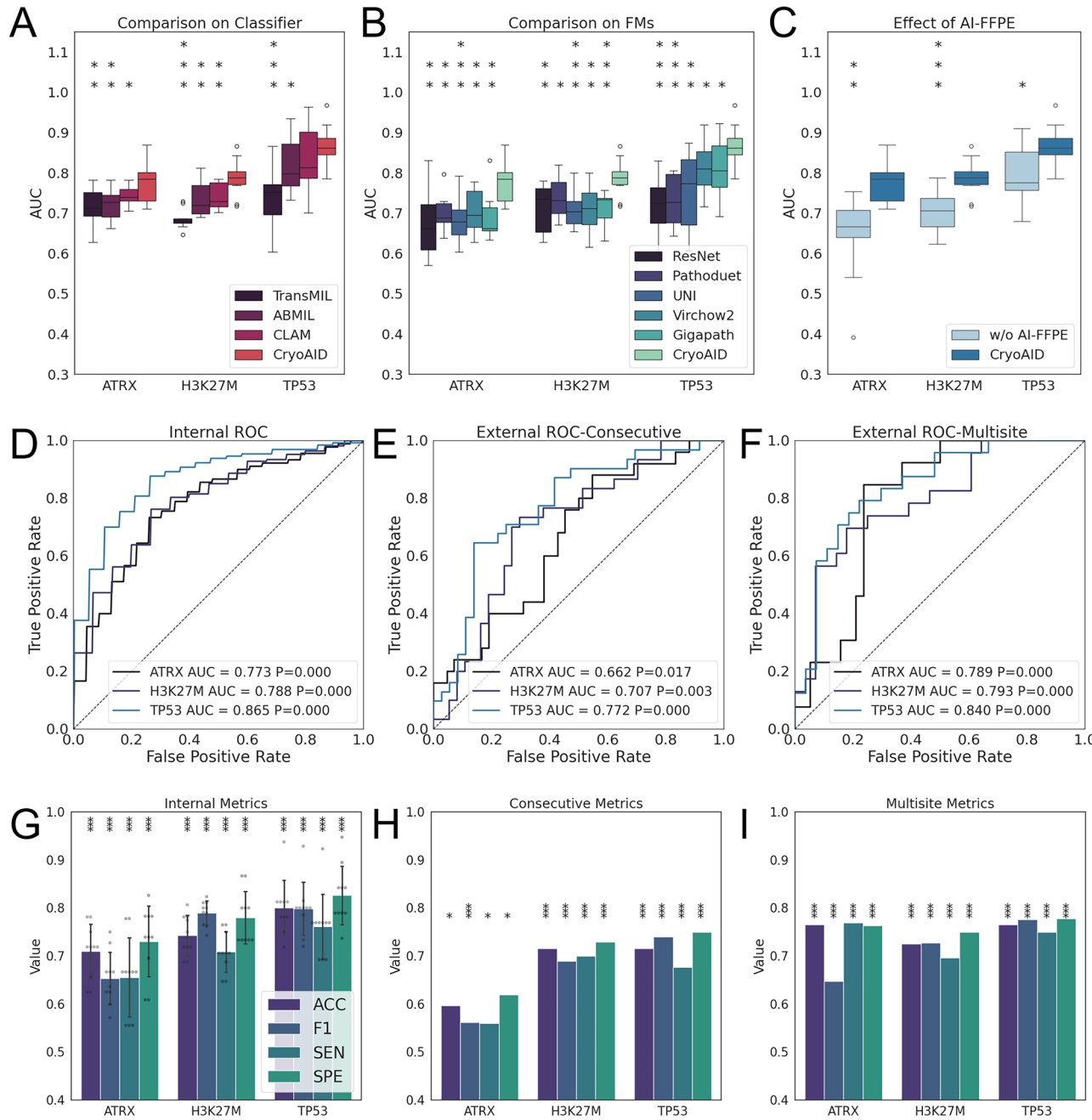

**Fig. 2 | CryoAID captures multiple mutant statuses in Pass images.**
**A** Distributions of obtained AUC from folds on predictions of ATRX, H3K27M, and TP53, using different classifiers. **B** AUC distributions for different mutant predictions using different foundation models (FMs). **C** AUC distributions without and with AI-FFPE generation. **A**–**C** For boxplots, the centre line indicates the median, the lower and upper limits of the box indicate the 25% and 75% percentiles, and whiskers give the maximum and minimum values. Outliers are indicated with "o". All boxplots in (**A**–**C**) are derived from results of the ten-fold cross-validations ($n = 10$). *: CryoAID shows significant improvement over the indicated methods at $p < 0.05$ level using one-sided t-test (not adjusted). **: $p < 0.01$. ***: $p < 0.001$. The significance asterisks should be read vertically. **D**–**F** The receiver operating characteristic (ROC) curves based on predictions of ATRX, H3K27M, and TP53, in

internal, consecutive, and multi-centre dataset, respectively. *P*-values are from one-sided permutation tests (not adjusted). **D** presents the averaged AUC over folds. For the multi-centre dataset, the predictions from Fujian Hospital and North Hospital are combined. Centre-specific results are offered in Fig. S2. **G**–**I** We report accuracy, F1 score, sensitivity, and specificity of CryoAID for different predictions, in the internal, consecutive, and multi-centre dataset respectively. ACC: accuracy. F1: F1 score. SEN: sensitivity. SPE: specificity. The metrics are estimated under the optimal threshold. Data in (**G**) are presented as mean +/− standard deviation STD ($n = 10$). Scatter in (**G**) depicts individual data from each fold ($n = 10$). *: Metric is significantly higher than chance level at $p < 0.05$ level using one-sided permutation tests (not adjusted). ***: $p < 0.001$. Source data are provided as a Source Data file.

surgeons more conservative, which creates a contradiction. To address this hurdle, we extend to propose a dual-validation protocol using CryoAID, where CryoAID offers both molecular diagnosis on key genes and histopathological diagnosis about low-grade tumours (Fig. 7A). We performed a five-fold cross-validation (8:2 train-test split)

with 301 subjects (33 pilocytic astrocytoma). Preliminary analysis suggests that CryoAID obtains above 80% ACC with a near 0.90 AUC on PA histopathological identification on Pass frozen sections (Fig. 7B, C). The F1 scores remain limited, potentially due to the small PA sample size for training. Given the hallmarked histopathological feature of PA,

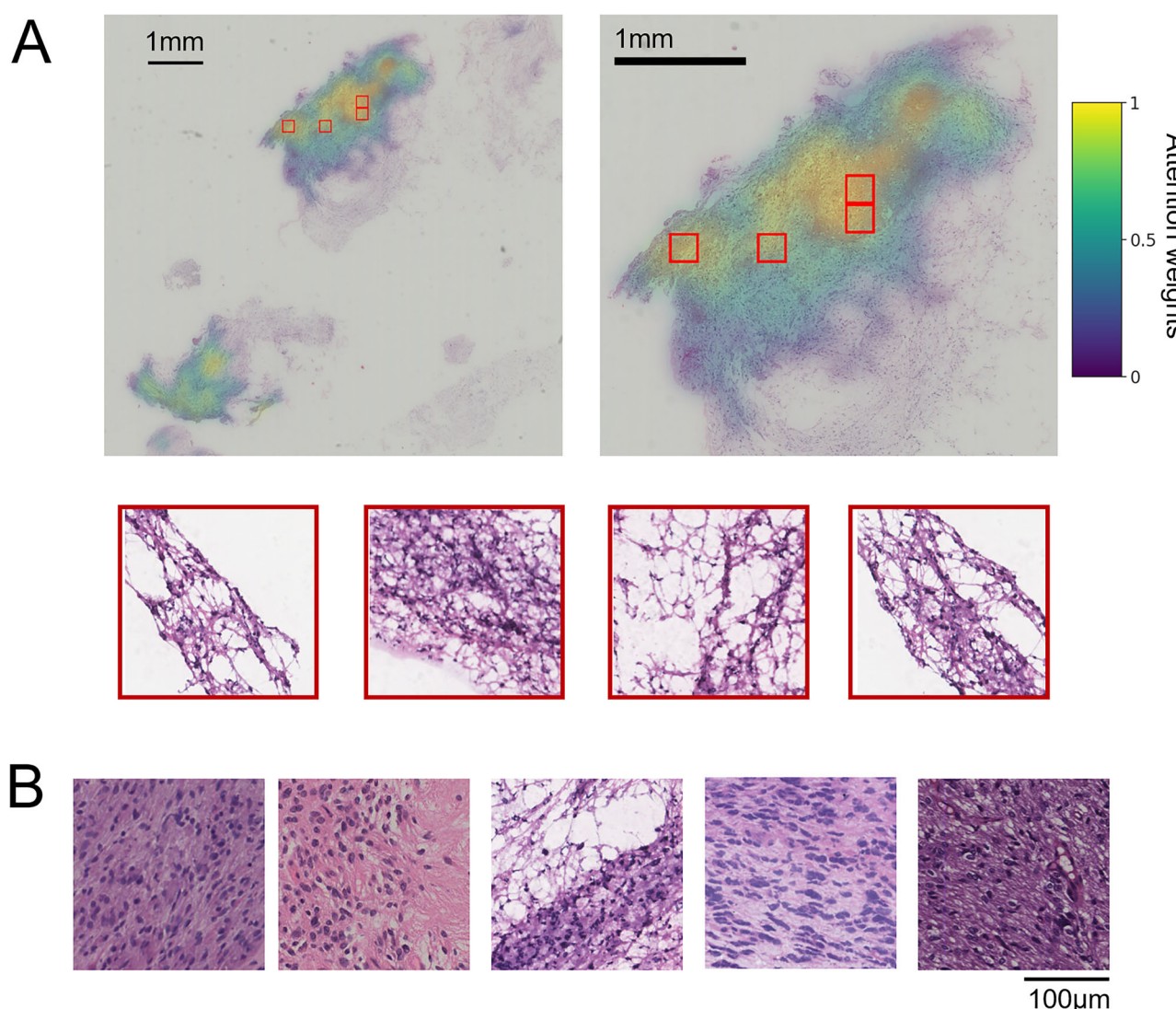

**Fig. 3 | CryoAID offers explainable features for the H3K27M mutant status prediction in Pass images. A, B** The exemplified slide and the corresponding heat map showing the spatial distribution of attention weights. The representative predictive patches from high attention regions are selected and attached. **B** Exemplified predictive patches from other patients.

i.e., Rosenthal fibers, the rechecking of a pathologist can confirm the potential diagnosis of PA, even upon a false positive prediction on H3K27M from CryoAID. This dual-validation procedure provides valuable guidance for neurosurgeons towards improved resection for full-spectrum midline gliomas.

## Discussion

Reducing the surgical risk of midline glioma is crucial for improving the quality of care and enhancing the efficiency of surgical workflow. In this study, we developed and evaluated an AI system, CryoAID, employing a cryosection-based approach for the molecular diagnosis of DMG. We demonstrated that CryoAID improves the prediction accuracy and workflow efficiency for genetic mutation detection from cryosection images across multiple centres. In particular, CryoAID exhibits a remarkable sensitivity in evaluating No-Pass cryosections, underscoring its potential to refine the decision-making protocols in routine biopsy operations. Our findings support the notion that CryoAID could potentially reduce the incidence of unnecessary No-Pass outcomes, shorten operation durations, and mitigate surgical hazards in DMG surgeries.

Most prior studies have focused on molecular assessment in FFPE samples[27–29] and those passed cryosection images[20]. While these results are promising, they overlooked the challenges of intraoperative diagnosis arising from the presence of No-Pass images. Furthermore, prior studies have not demonstrated whether such models can generalize in a real-world surgical deployment to support efficient clinical decision-making. In this study, we provide a robust, multi-centre evidence of AI performance across both Passed and No-Pass cryosectioned cryosection images. The emphasis on AI-assisted intraoperative diagnosis distinguishes our study from prior works[20,27]. In the consecutive data evaluation (Fig. 6G), CryoAID saved 38 biopsy attempts out of 143 times in the 68 patients, demonstrating substantially greater efficiency than conventional human-based workflows. This improved efficiency allows clinicians to minimize repeat biopsies—an important advancement for high-stakes DMG treatment and patient care.

CryoAID offers opportunities to repurpose specimens previously deemed non-viable in traditional histopathology. We offered key evidence to support that tumour's genetic information could be recovered from No-Pass cryosectioned images even when tumour cells are visually obscured (Figs. 4, 5). Key molecular alterations in midline

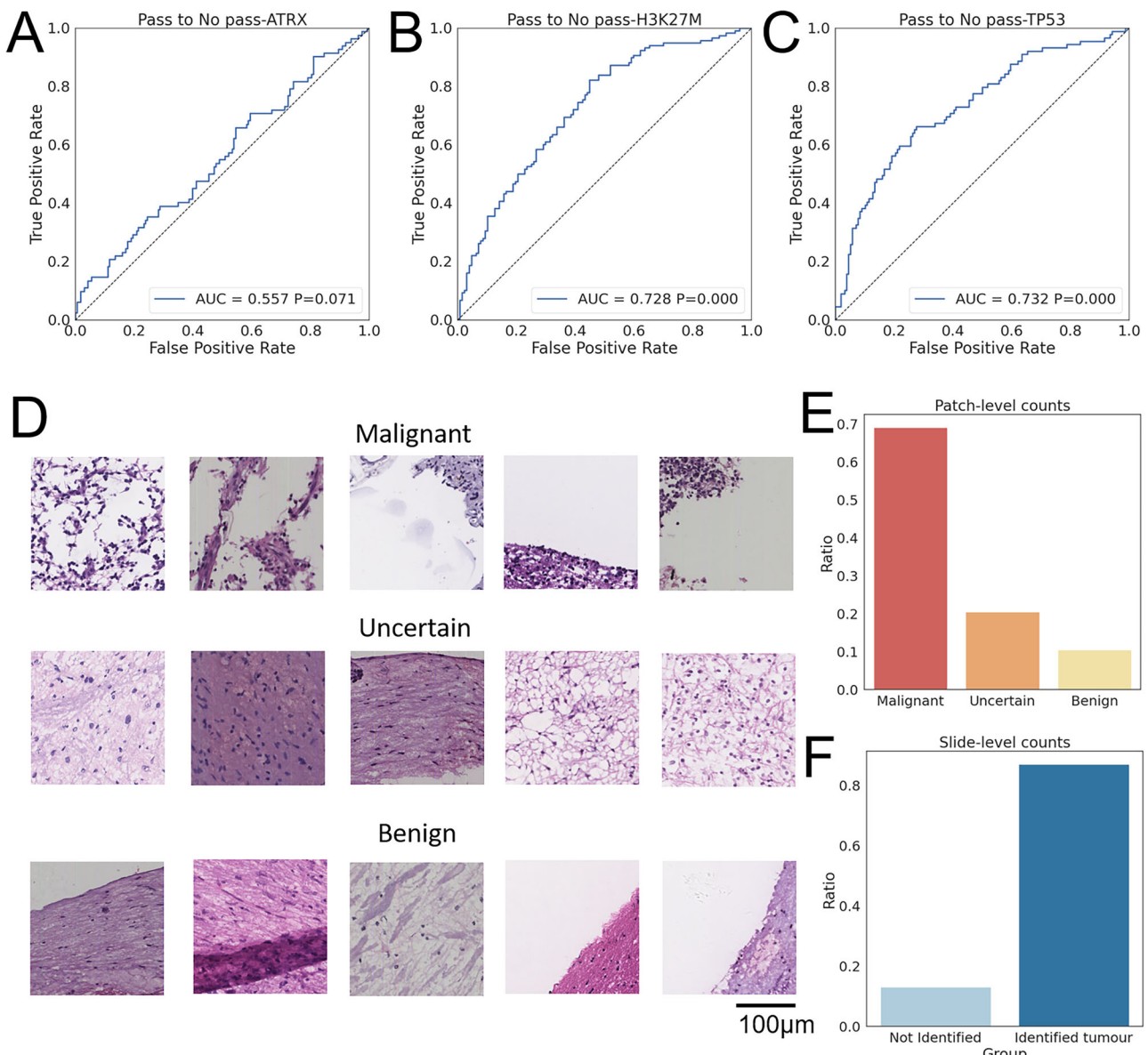

**Fig. 4 | CryoAID trained on Pass images detects mutant status in No-Pass images. A–C** The ROC curves from Pass image pre-trained CryoAID on predictions of ATRX, H3K27M, and TP53 in No-Pass images, respectively. *P*-values are associated with the AUC values, yielded from permutation tests. **D** The exemplified top predictive patches from accurately mutant-identified No-Pass images grouped into malignant, uncertain, and benign patches based on visual checking of pathologists. **E** Bar plots for the distributions of counts of the grouped top predictive patches. **F** Slide-level counts of slides with and without tumour cells identified. Source data are provided as a Source Data file.

gliomas, such as the H3K27M mutation, have become definitive criteria for pathological diagnosis[31]. However, the lengthy clinical process of FFPE, IHC, and genetic testing hinders rapid intraoperative molecular interpretation. To respond, the CryoAID model directly extracts the genetically related image features from both Pass and No-Pass cryosection images. Moreover, Moreover, growing evidence links tumour genotypes to morphological transformations in peritumoural cells[17,19], suggesting that mutated tumour cells may induce distinct morphological signatures in adjacent tissues[18]. In our study, CryoAID revealed explainable diagnostic features, including suspected tumour cells, glial cell proliferation, abundant cytoplasm, and localised endothelial cell proliferation (Fig. 5). This key finding reflects that CryoAID is able to discern peritumoural cues that elude human detection to infer the challenging molecular-level information[32,33].

A key contribution of our study lies in leveraging the generative power of AI to extend the scope of pathological foundation models,

typically pre-trained on large-scale FFPE samples. Unlike prior studies demonstrating their validity in FFPE-based tasks[23–26], we found that directly transferring a pathological foundation model to cryosection images yielded suboptimal performance (Fig. 2C). In our study, the state-of-the-art AI-FFPE module introduces a generation-based domain adaptation strategy[34,35], preserving the performance of pre-trained models while avoiding the need for extensive fine-tuning on cryosection datasets. The AI-FFPE generative model effectively transforms cryosection images into FFPE analogues, bridging the imaging gap and enabling high-fidelity feature extraction. The success of this generative approach is further reflected in the model's ability to generalize to other tasks. For instance, through the integration of AI-FFPE and CHIEF models, our framework demonstrates strong adaptability, requiring only minimal Transformer fine-tuning to analyse other tumour types (such as PA) that demand intraoperative diagnostics.

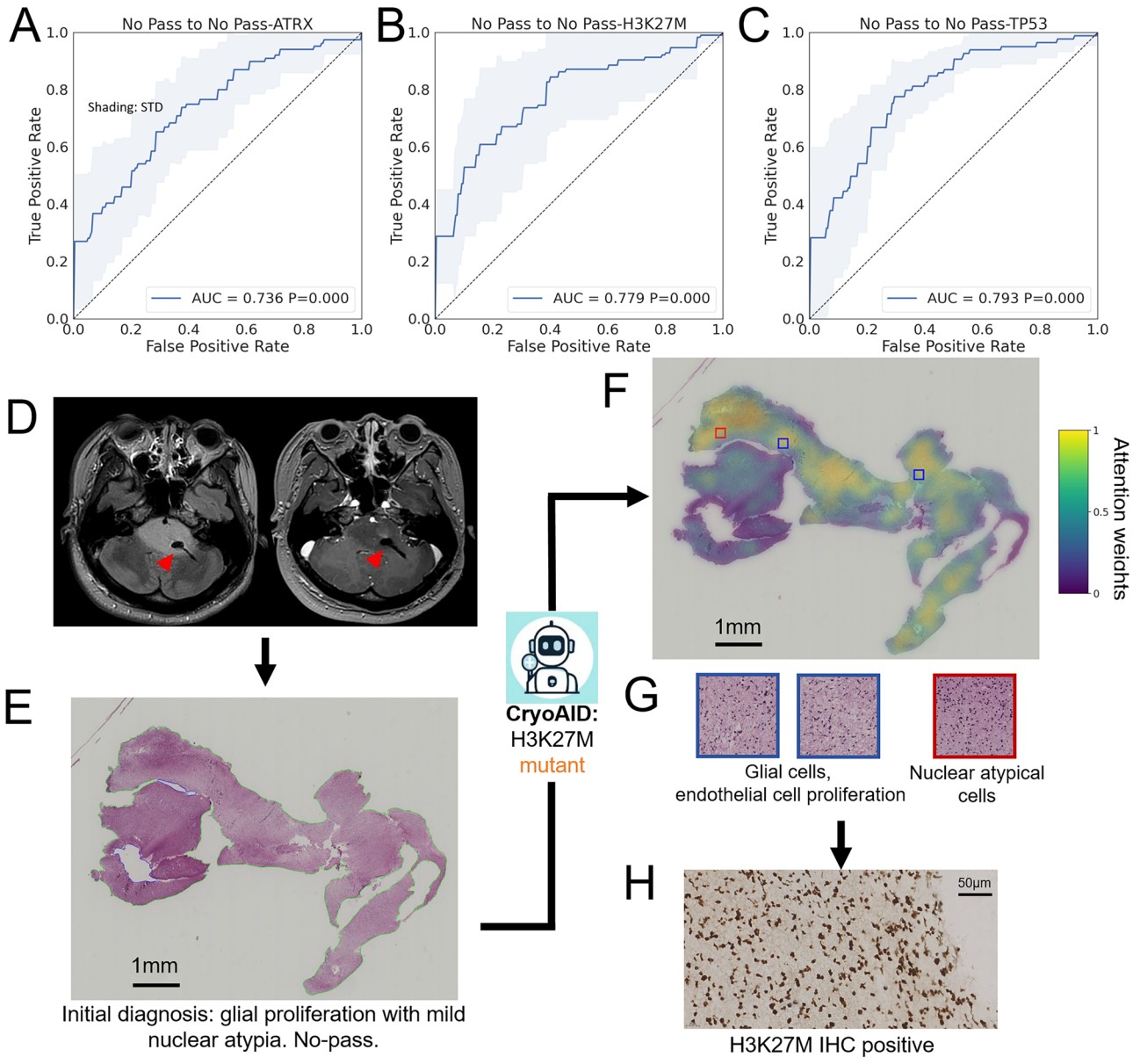

**Fig. 5 | CryoAID learns on No-Pass images to assess mutant status. A–C** The averaged ROC curves from CryoAID trained on No-Pass images on predictions of ATRX, H3K27M, and TP53 in No-Pass images, respectively. The *p*-values are associated with the AUC values, yielded from one-sided permutation tests (not adjusted). The shading represents the STD over folds. **D** Biopsy positions in the magnetic resonance image, highlighted by red arrows. **E, F** The exemplified slide and the corresponding heat map showing the spatial distribution of attention weights. **G** The exemplified predictive patches. **H** IHC results using cryosection samples in (**E**), supporting a H3K27M mutant. Source data are provided as a Source Data file.

The development of CryoAID aligns with the advances of Fas-tGlioma model[36], which established a foundation model based on stimulated Raman histology (SRH) to facilitate intraoperative detection of glioma infiltration and reduce the duration of surgical procedures. While both approaches aim to accelerate intraoperative workflows, CryoAID uniquely focuses on reducing unnecessary biopsies and mitigating surgical risks and patient morbidity. In particular, CryoAID provides a dual-validation by considering both molecular and histopathology assessments, thereby supporting multi-dimensional intraoperative decision-making. The clinical utility of FastGlioma is limited to the availability of SRH imaging systems, whereas CryoAID is designed to operate seamlessly within the routine neurosurgical workflow using standard cryosection images. This compatibility with existing diagnostic infrastructure greatly enhances CryoAID's translational potential across diverse clinical environments.

Our study has several limitations that warrant further investigation. Owing to limited sample sizes, BRAF alterations (including V600E mutations and KIAA1549–BRAF fusions) were analyzed as a unified category without subclassification. Similarly, other molecular alterations, such as IDH mutations and MYCN amplifications, were excluded due to insufficient representation. Future multi-centre collaborations could validate CryoAID's performance through expanded genomic profiling efforts. Additionally, our current approach may not benefit extremely challenging cases requiring multiple repeated biopsies. Enhancements may involve incorporating a broader spectrum of molecular markers and implementing dual-validation strategies from both histological and genetic perspectives. Finally, as this study was conducted retrospectively without pathologist involvement, a prospective clinical trial integrating AI–pathologist collaboration is warranted to assess CryoAID's real-world robustness in reducing biopsy frequency.

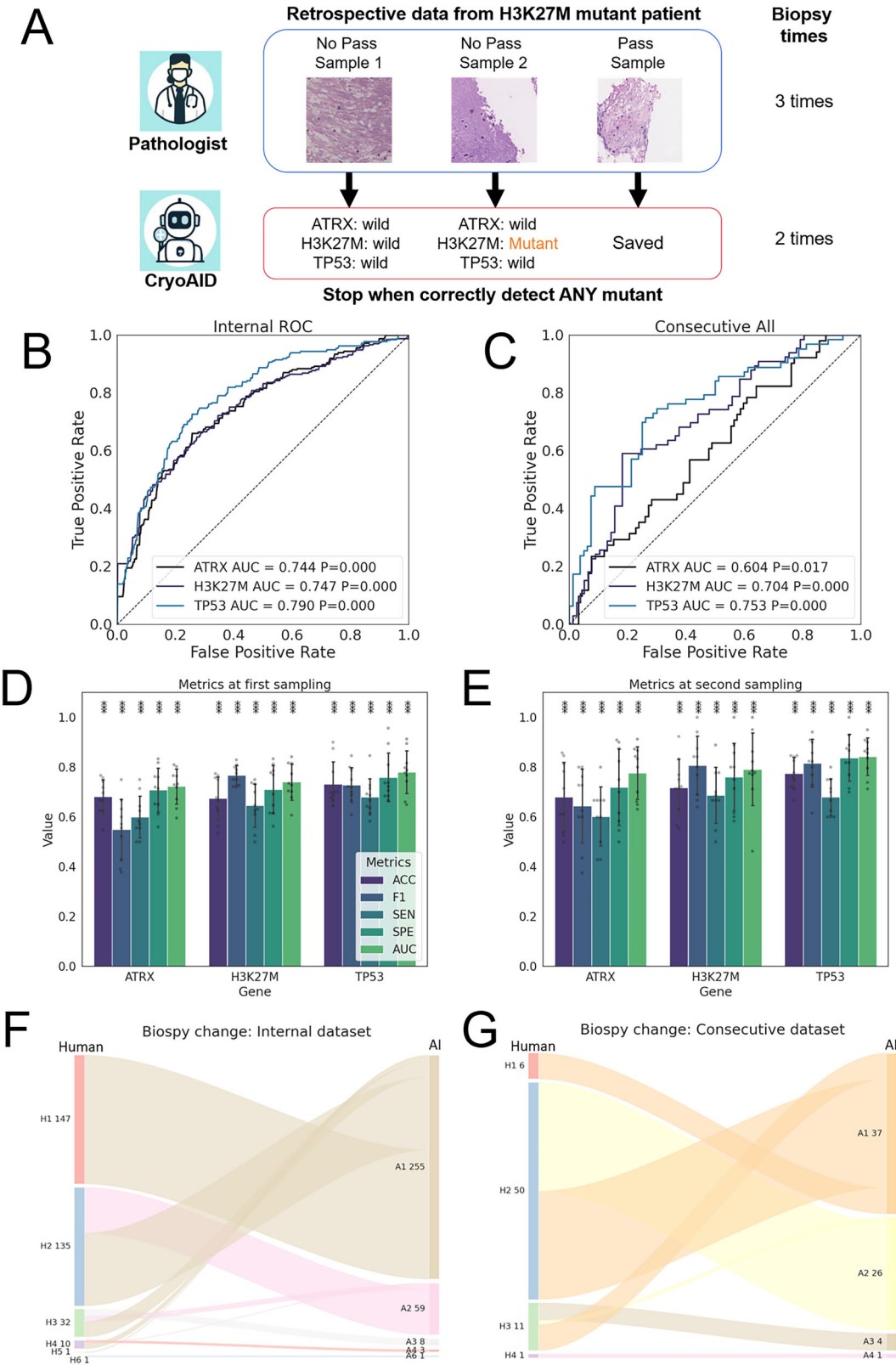

In conclusion, this study demonstrates the real-time value of AI-augmented workflows in reducing re-biopsy efforts during DMG surgeries. Our findings underscore the clinical utility of cryosection pathology and its explainable contribution to informed neurosurgical decision-making. By leveraging the strengths of foundation models, CryoAID has the potential to improve diagnostic efficiency, enhance the quality of care for patients with DMG, and accelerate the broader integration of AI into intraoperative surgical practice.

## Methods

### Sample acquisition and pathological diagnosis

In this retrospective study, the data inclusion and exclusion are illustrated in Fig. 1C and S1. The cryosection whole slide image (WSI) from

**Fig. 6 | CryoAID trained on both Pass and No-Pass images can assist DMG biopsy decision making. A** An illustration of the retrospective biopsy re-checking scheme. **B** The averaged ROC curves for ATRX, H3K27M, and TP53 predictions using CryoAID in all data (both Pass and No-Pass images) of internal datasets. **C** Corresponding ROC curves in all data of consecutive testing dataset. In **B**, **C**, $p$-values are associated with the AUC values, yielded from one-sided permutation tests (not adjusted). **D**, **E** The accuracy (ACC), F1 score, sensitivity (SEN), specificity (SPE) and AUC for model's prediction, given the first two times of biopsies in the internal dataset. Data are presented as mean +/− STD ($n = 10$). Scatters show individual data from each fold ($n = 10$). ***: Metric is significantly higher than chance level at $p < 0.001$ level using one-sided permutation tests (not adjusted). **F**, **G** The Sankey diagrams for the changes of biopsy counts to accurately detect any mutant by human and AI in internal and consecutive testing datasets. The flow between category bars indicates the corresponding changes. The format "Hn m" means human successes with n times biopsies has m samples. "An m" offers the corresponding data from CryoAID. Source data are provided as a Source Data file.

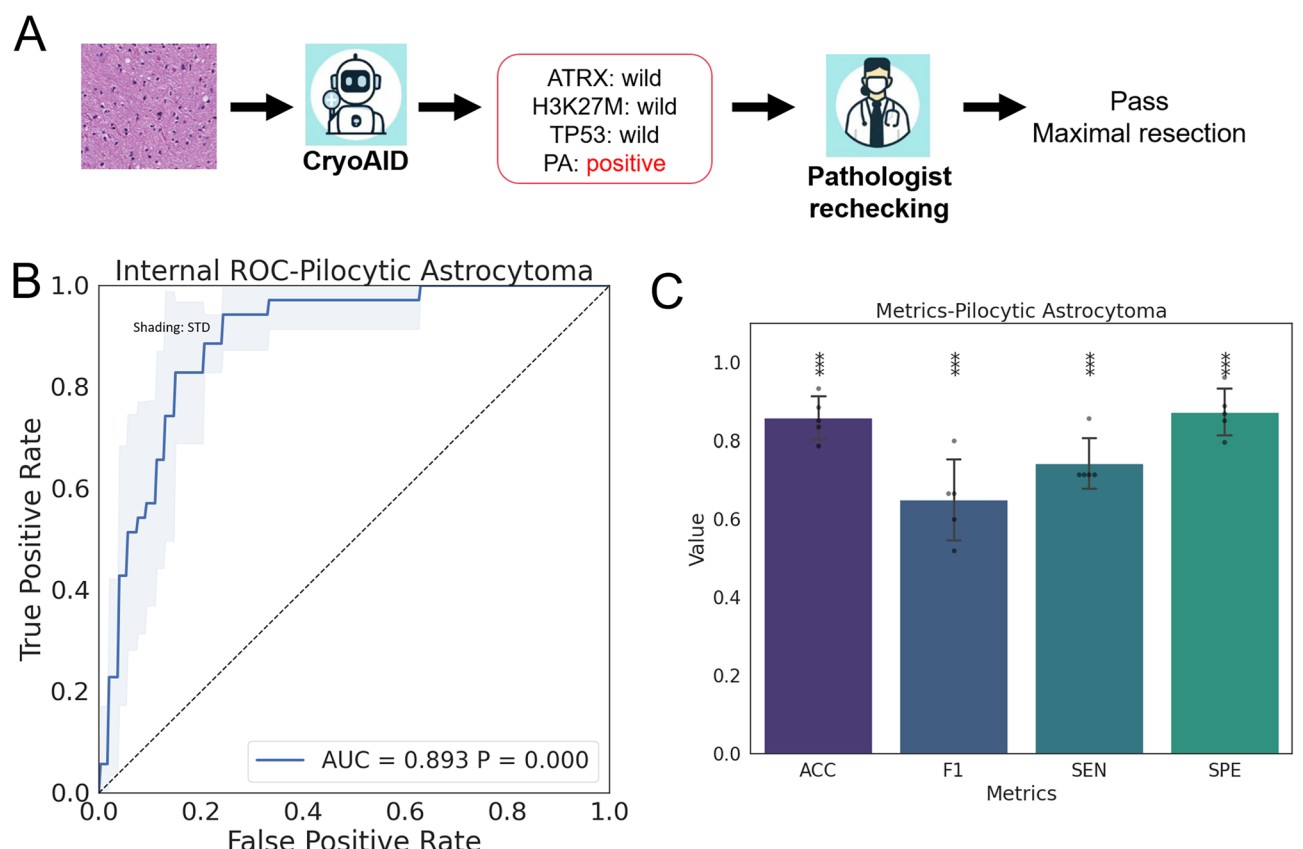

**Fig. 7 | Diagnostic performance of CryoAID on pilocytic astrocytoma.**
**A** Workflow of dual validation. PA: pilocytic astrocytoma. **B** Averaged ROC curves over folds. Shading indicates the STD. **C** Averaged prediction metrics. Data are presented as mean +/− STD ($n = 5$). Error bar indicates the standard deviation. Scatter offers individual data. ***: Metric is significantly higher than chance level at $p < 0.001$ level using one-sided permutation tests (not adjusted). Source data are provided as a Source Data file.

Huashan main campus ($n = 107$), west campus ($n = 202$), and Shanghai Gamma Hospital ($n = 17$) were used for internal cross-validation (cases were retrospectively collected from January 2018 to December 2022). There were a total of 564 WSIs from 326 patients (238 No-Pass images). After the model was established, we used the consecutive dataset (cases enrolled after January 2023), containing 143 WSIs from 68 patients (75 No-Pass images), which was collected from the west campus of Huashan for a consecutive validation. In addition, the Pass images from the First Affiliated Hospital of Fujian Medical University ($n = 17$) and the north campus ($n = 35$) were collected for the multi-centre external validation. The demographics information and mutant information for each dataset are shown in Table S1. Sex was self-reported by participants and included in the demographics information analysis.

All imaging sequences were imported into the neuro-navigation system (Medtronic S7, Minneapolis, MN, USA) and co-registered with preoperative magnetic resonance images. For biopsy surgery, targets were carefully chosen and marked by senior neurosurgeons (10-year experience). After craniotomy and before a complete dural opening,

cylindrical samples were obtained under conventional stereotactic biopsy procedures. For resection surgery, after craniotomy and dural opening, suspicious tissues were selected and sampled for further cryosection pathologic diagnosis. The pathologic assessment of the cryosection slides was performed by two senior pathologists independently. The genetic mutant status was determined using IHC.

**Whole slide imaging acquisition and pre-processing**
We preprocessed the cryosection WSI using the protocol proposed in Lu et al. [37], which includes tissue segmentation and patching. The WSI was initially downscaled by 20× and converted from read-green-blue (RGB) to hue-saturation-value (HSV) colour space. Then a binary mask was created for tissue areas by thresholding the saturation channel and smoothing edges with median blurring, followed by morphological closing to eliminate small gaps. Only the foreground objects that meet a specific area threshold are kept for further analysis, and a segmentation mask is produced for each slide. In this study, we directly adopted the "biopsy" configuration provided by Lu et al. [37] to set the hyperparameters for the segmentation algorithm. Post-segmentation,

patches with a size of 224 × 224 pixels were extracted from these foreground areas at a 20× magnification. To refine the quality of segmentation results, we further filtered out image patches that are contaminated (e.g., black regions, manual markers, or dust) or fail to show segmented tissues (e.g., blank regions) by applying the RGB thresholds. The filtered results were sampled and double-checked visually.

## CryoAID

**AI-FFPE**. Towards strong feature extraction on cryosection images, we applied the AI-FFPE method[22] to transfer cryosection patches to FFPE-like patches. AI-FFPE is a generative adversarial network incorporating an attention mechanism and a self-regularisation constraint, namely Contrastive Unpaired Translation (CUT)[38]. In particular, it can rectify the cryosection artefacts and change the appearance of cryosection images similar to real FFPE images. In the whole process, the clinically relevant visual features, such as the cell structure, are preserved for quantitative analysis. Exemplified pairs of inputted cryosection images and generated AI-FFPE images are shown in Figs. 1B and S3. Note that the generation will not introduce new artefacts.

**Foundation model for feature extraction**. All AI-FFPE processed patches were fed into a pre-trained pathology foundation model, called CHIEF[39] (Clinical Histopathology Imaging Evaluation Foundation), for the feature extraction. CHIEF is a pathology foundation model developed to enhance cancer diagnosis and prognosis prediction. Trained on 15 million unlabeled image patches and over 60,000 whole-slide images across 19 cancer types, CHIEF employed self-supervised and weakly supervised learning to extract robust features from histopathology images. It demonstrates strong accuracy in tasks of cancer detection, tumour origin identification, genomic mutation prediction, and survival prognosis. We thus leveraged CHIEF's information-extraction ability in our tasks of multiple molecular diagnoses.

**Transformer classifier**. After feature extraction, we utilised a Transformer architecture[40] to build a classifier model to predict the mutant status of patients. A Transformer block consists of a self-attention layer, a multi-layer perceptron (MLP) layer, and skip connections. The self-attention layer enables the model to weigh the importance of different patches. Differing to conventional attention, self-attention computes the importance based on both individual patch features and their relationships. Following the self-attention layer, the output is passed through a layer normalisation step and then an MLP layer. The MLP consists of two fully-connected layers with a ReLU activation in between, allowing the model to learn complex mappings from the self-attention outputs. We also employ skip connections (known as residual connections) around both the self-attention and MLP layers. We use the skip connection as it helps mitigate the vanishing gradient problem by allowing the gradient to flow directly through the network. Layer normalisation is then applied to stabilise and accelerate the training process. We denote the inputted patches as $x$, and $N_x$ as the number of inputted patches. Formally, the process is expressed as:

$$K(x) = W_K x,\ Q(x) = W_Q x,\ V(x) = W_V x, \tag{1}$$

$$A = \mathrm{softmax}\left((K(x))^T Q(x)\right), \tag{2}$$

$$y_{sa} = x + \mathrm{MLP}(AV(x)), \tag{3}$$

$$y_{MLP} = y_{sa} + \mathrm{MLP}(y_{sa}), \tag{4}$$

where $A$ is a $N_x$-by-$N_x$ self-attention weight matrix, $y_{sa}$ is the intermediate output from self-attention layer, and $y_{MLP}$ is the output from the MLP layer in the Transformer block. Note that we did not implement positional embedding and relied on the feature representation from the used foundation model.

**Feature visualisation**. The self-attention layer in the Transformer enables an exploration of the importance of each patch in a slide during the prediction. Following Qu et al. [41], based on Eqs. (2) and (3), we regarded the averaged logarithmic value for each column in $A$ as a rough estimation for the weight on each patch. Denote $W_i$ is the importance weight on the $i$th patch,

$$W_i = \frac{1}{N_x} \sum_j^{N_x} \log\left(A_{ij}\right). \tag{5}$$

We analysed patches with top weights in a slide and utilised representative patches for visualisations. The weights were also mapped back to the original location to produce the "heat map" for checking the spatial distribution of features.

**Implementation**. We applied the pre-trained parameters of the AI-FFPE model (trained on brain tumour images) and the CHIEF model without further tuning. In the classifier implementation, after the processing of one Transformer block (one-head self-attention, attention encoding dimension = 64, MLP dimension = 1024), an average pooling is utilised to integrate the patch-level feature into the WSI-level feature. And one MLP (input dimension = 768, output dimension=2) with Softmax following the Transformer block works on the WSI-level feature to generate the prediction probability.

A ten-fold cross-validation was implemented in the internal dataset with a train-vs-test split ratio of 9:1 for each round. We trained the Transformer classifier with training epochs = 80, learning rate = 1e-4, and batch size = 1. We used the Adam algorithm to automatically optimise the trainable parameters with a weight decay of 5e−3. A weighted sampler was used to tackle the issue of class imbalance. Cross-entropy was used as the loss function. Other parameters of the neural network models were initialised with random weights. The training was accelerated with a single Nvidia GTX 4090 GPU.

For the external testing, we trained a new model based on the internal cross-validation dataset and applied the new model to the target testing dataset. The prediction models for different genes were independently established. In the Pass-to-No-Pass analysis (Fig. 4), the ten models trained on internal Pass images during cross-validations were directly applied to all internal No-Pass images. A probability-level majority voting was conducted by averaging the prediction probabilities from the ten models, resulting in the final prediction results. For feature visualization in Fig. 4, one of the ten models showing the highest AUC in No-Pass images was selected to generate the predictive patches.

## Comparison methods

To assess the effectiveness of each component in CryoAID, we systematically compared three classes of competing methods in terms of classifier, foundation model, and generative ability. All methods were evaluated using a consistent training and validation pipeline with hyperparameters tuned on the same validation splits to ensure fair comparison. We compared all methods using the internal cross-validation dataset, as it represents the large-scale collected data.

First, we compared different choices of classifiers, within attention-based multiple instance learning (ABMIL)[42], CLAM[37], TransMIL[43], and Transformer (CryoAID). ABMIL classifier learns to select disease-related instances in a given bag of batches via an attention mechanism for classification. CLAM classifier is based on

conventional ABMIL, with the aid of a support vector machine loss to increase the discrepancy of the embedded feature of two classes. TransMIL combines MIL and Vision Transformer. The cross-talk among patch features is introduced using a Vision Transformer, and the integrative "cls_token" over all instance features is used for classification. However, the increased TransMIL model complexity can result in overfitting under small training samples.

Second, we compared pathology foundation models in CryoAID, including ResNet50, UNI[23], Pathoduet[26], Virchow2[24], Gigapath[25], and CHIEF[39] (CryoAID). ResNet50 was trained on large-scale natural images rather than pathology images. UNI and Pathoduet were trained on roughly 100 K WSIs and focused on visual tasks such as segmentation and tumour detection. Gigapath and Virchow2 were respectively trained on 170 K and 3,1 M WSIs and demonstrated the scalability of the foundation model for real-world applications, including rare tumour diagnosis.

Finally, we made an effort to skip the AI-FFPE step in CryoAID to assess the effectiveness of using generative-based image enhancement of CryoAID.

### Statistical analysis

**Model comparison.** To evaluate the model's performance on the internal cross-validation and external test, we used the area under the ROC curve (AUC). The ROC curve was created by plotting the true positive rate against the false positive rate at various decision probability thresholds. AUC informs the capability of a model in distinguishing between two classes. Additionally, we computed the accuracy (ACC), sensitivity (SEN), and specificity (SPE), based on the optimal decision probability threshold on the ROC curve, achieving the balance between SEN and SPE. F1 score is also reported, yielded from a separate F1-oriented threshold searching. Two distinct thresholds were used because the optimal operating point for maximizing F1 often differs from that for balancing sensitivity and specificity, reflecting different trade-offs between false positives and false negatives. For models from different folds or different centres, the thresholds were estimated separately, and the thresholded decision values were combined. To compare CryoAID with different models, we applied a one-sided two-sample t-test on these AUCs from folds and assessed the significance (P-value) of the improvement. In the internal cross-validation, we reported the mean, standard deviation (STD), and 95% confidence interval (CI) from the folds. For AUC based on one-time testing, the CI is estimated using DeLong's method. In addition, the significance of the AUC score was estimated by 1000 permutations on the truth label to estimate the distribution of the AUC under the chance level. For cross-validation-based results, including ACC, SEN, SPE, and AUCs, the permutation test is performed based on the integrated predictions and labels over all folds.

**Retrospective biopsy re-checking.** The AUCs of the model's prediction given once and twice biopsies were both calculated. To estimate the prediction probability upon twice biopsies for AUC computation, we averaged the probabilities of the model's prediction on the first and the second sampling. The corresponding ACC, SEN, and SPE were also computed by the optimal cut-off on the ROC curve as mentioned above. This is to demonstrate the model's capacity to detect mutant samples while making the least-mistaken recall in normal samples. The biopsy counts for achieving molecular diagnosis by human (pathologists) were obtained as the real counts of biopsies to obtain the Pass cryosectioned image. The biopsy counts for AI were determined by the order of the first image in the sequence, where CryoAID accurately predicted any mutant. This statistic indicates the potential maximum reduction in re-biopsies when using CryoAID, while practical implementation relies on a proper collaboration between CryoAID and the pathologist. The statistical comparison for the biopsy counts was performed using a one-sided paired t-test.

### Ethical approval

This research was approved by the Institutional Review Board of Huashan Hospital (KY2024-1242) and was registered in the Chinese Clinical Registry (No. ChiCTR2400093046). Patients have signed informed consent forms for enrolling in the CNS disease bank in advance of operations, authorizing to use their pathological images for this study.

### Reporting summary

Further information on research design is available in the Nature Portfolio Reporting Summary linked to this article.

### Data availability

The multi-centre clinical dataset (extracted pathology image feature and gene mutant label, being anonymous, together with trained model and data splits) generated in this study has been deposited in the Zenodo database under accession code 17373753 [DOI: 10.5281/zenodo.17373753]. The data are available under restricted access for the privacy protection restriction of the hospital; access can be obtained by sending an online request via Zenodo. The request will be reviewed online and responded to within one week. The approval will be given based on the reasonableness of the research purposes of the request, and commercial usage is strictly forbidden. Source data are provided with this paper.

### Code availability

We used Python 3.10 and PyTorch 2.2.2. For previously published models and methods: The model for AI-FFPE is available at https://github.com/DeepMIALab/AI-FFPE. The CHIEF model is available at https://github.com/hms-dbmi/CHIEF. UNI model is available at https://github.com/mahmoodlab/UNI. The Gigapath model is available at https://github.com/prov-gigapath/prov-gigapath. Virchow2 model is available at https://huggingface.co/paige-ai/Virchow2. The PathoDuet model is available at https://github.com/openmedlab/PathoDuet. The code for ABMIL is available at https://github.com/AMLab-Amsterdam/AttentionDeepMIL. The code for CLAM and the ResNet model is available at https://github.com/mahmoodlab/CLAM. The code for TransMIL is available at https://github.com/szc19990412/TransMIL. Our self-developed codes are documented at https://github.com/MianxinLiu/CryoAID (https://doi.org/10.5281/zenodo.17393827)[44] and released under MIT license.

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

## Acknowledgements

This study is supported in part by National Key Research and Development Program of China, MOST (No. 2023YFC2510000 to YM), Non-communicable Chronic Diseases-National Science and Technology Major Project (No. 2023ZD0511800 to YFY), National Natural Science Foundation of China (No. 82272063, 82127801 and 82227806 to QY), Shanghai Urban Digital Transformation Specializing Fund (No. 202301071 to YM), Shanghai Hospital Development Centre (Clinical Research Plan of SHDC 2022CRW004 and Clinical Competence Improvement and Advancement in Neurosurgery plan of SHDC22024312 to YM), the Fujian Province Science and Technology Innovation Joint Fund (No. 2021Y9135 to JPS), AI4S Climber Initiative, Intern Discovery, and Shanghai Artificial Intelligence laboratory (CFS, LB, and MXL).

## Author contributions

Conceptualization, Y.F.Y., Q.Y., M.Z., M.X.L. and Y.M.; Methodology, Y.F., M.Z. and M.X.L.; Software, Y.F., K.T.C., and M.X.L.; Validation, Y.F.Y. and M.X.L.; Data interpretation, Z.G.D., X.F.W., and H.X.C.; Resources, Z.G.D.,

X.F.W., H.X.C., J.P.S., C.F.S., L.B., Y.M.; Data curation, Y.F.Y., Y.N.W., Z.G.D., X.F.W., H.X.C., Z.Y.W., J.P.S.; Writing—original draft preparation, Y.F.Y. and M.X.L.; Writing—review and editing, Y.F.Y., Q.Y., M.Z., M.X.L. and Y.M.; Visualization, Y.F.Y., and M.X.L.; Supervision, M.Z., M.X.L., Y.M.; Funding acquisition, Y.F.Y., Q.Y., J.P.S., C.F.S., L.B., M.X.L., and Y.M.; All authors have read and agreed to the published version of the manuscript. All authors contributed to the article and approved the submitted version.

## Competing interests

The authors declare no competing interests.

## Additional information

Qi Yue, Mu Zhou, Mianxin Liu or Ying Mao.

