## [Peer Review File · Nature Communications]

AI-augmented Intraoperative Decision-making Workflows in Diffuse Midline Glioma Biopsy using Cryosection Pathology

Corresponding Author: Dr Mianxin Liu

Version 0:

Reviewer comments:

Reviewer #1

(Remarks to the Author)

The authors have adequately responded to my critiques and the criticisms of the other reviewers.

(Remarks on code availability)

The tutorial for the code provided is lacking. It would increase likely adoption of the method if a more clear example of usage, expected inputs and outputs, etc. were provided.

Reviewer #2

(Remarks to the Author)

The authors have substantially improved the manuscript over the original submission, addressing many of my concerns. I have only a few additional comments related to issues that still need clarification or correction.

1. The authors fail to present the F1 score as requested in revised manuscript. This is an important metric, especially in datasets that likely have substantial class imbalance. They already have presented most of the necessary data to calculate these values and it would be useful if they were included.
2. The numbers do not seem to be correct in Figure S1. 652 imaging highly suspected patients - 142 excluded due to pathology excluded glioma = 510 not 457 as reported; After excluding patients excluded due to lack of frozen pathology (n=63) the number 447 seems it would be correct if subtracted from 510.
3. In previous review Q7, an output on the ensemble was requested. The authors indicated that was already being deployed and depicted in Figure 6. While I believe that I understand how this reduces biopsy times by the author's assertion, it also seems to be slightly confusing as presented in the Figure. Specifically, Figure 6a indicates 'stop when correctly detect any mutant'. This approach would confound the results and overemphasize the performance metrics. In reality you would not have the correctness or incorrectness to rely upon. Therefore, the biopsy would stop when any mutation was detected (right or wrong) and then you would determine the correctness of the model after the fact. If the stopping rules are dependent on the correctness, it seems it would exclude false positive examples increasing both the accuracy and the specificity.

Still quite a few copy editing errors:

Examples

1. Abstract:

Cryosection pathology is essential for intraoperative diagnosis of diffuse midline gliomas, yet it often leads to diagnostic errors and prompt unnecessary re-biopsies prior to the formal molecular assessment can be completed.

2. Flow chart in extended data '652 imaging highly suspected patients were remained' (and other mentions of 'were remained')
3. 'PAs 341 require maximal resection while an indication of H3K27M mutant could make surgeons to be conservative'
4. ..."CryoAID well detects the statuses in TP53"

There are quite a few additional grammatical errors that still make the manuscript difficult to follow.

(Remarks on code availability)

Reviewer #3

(Remarks to the Author)

1. Thank you for responding to the comments. Based on the authors' clarification, there does not appear to have been any misunderstanding in the summary section of my earlier remarks. The authors confirmed that the intended goal was in line with predicting "the mutation statuses of ATRX, H3K27M, and P53 using cryosection samples." This remark simply summarized the reported performance of such predictions in "Pass" images ("areas under the receiver operating characteristic curve (AUCs) ranging from 0.704 to 0.790") and "No Pass" images ("AUCs between 0.561 and 0.635") as presented in the original manuscript.
2. The discussion on "biopsy times" appears to have been copied and pasted from the response to Reviewer 2 and was not related to my comments. That said, this point was not central to understanding the authors' main message.
3. I am glad that the suggestions regarding foundation models contributed to substantial performance improvements and that the benchmarking analyses were useful in highlighting the strengths of the proposed approach.
4. The authors have adequately addressed the other previously raised comments. Thank you.

(Remarks on code availability)

The code provides a README file with sufficient instructions for installing the application.

REVIEWERS' COMMENTS

Reviewer #1 (Remarks to the Author):

The authors have adequately responded to my critiques and the criticisms of the other reviewers.

Reviewer #1 (Remarks on code availability):

The tutorial for the code provided is lacking. It would increase likely adoption of the method if a more clear example of usage, expected inputs and outputs, etc. were provided.

Thank you for your kind consideration and positive evaluation of our work. We have now deposited the data features, labels, trained models with data splits in Zenodo. In Github, we provide a more clear README file for users to follow and adapt the proposed method. We will further keep the Github page updated.

Reviewer #2 (Remarks to the Author):

The authors have substantially improved the manuscript over the original submission, addressing many of my concerns. I have only a few additional comments related to issues that still need clarification or correction.

1. The authors fail to present the F1 score as requested in revised manuscript. This is an important metric, especially in datasets that likely have substantial class imbalance. They already have presented most of the necessary data to calculate these values and it would be useful if they were included.

We have added F1 score into the related figures (see updated Figures 2, 6, 7).

2. The numbers do not seem to be correct in Figure S1. 652 imaging highly suspected patients - 142 excluded due to pathology excluded glioma = 510 not 457 as reported; After excluding patients excluded due to lack of frozen pathology (n= 63) the number 447 seems it would be correct if subtracted from 510.

Figure S1. The workflow of data inclusion and exclusion criteria.

We have checked and confirmed the correct number. Thanks for pointing out this calculation issue.

3. In previous review Q7, an output on the ensemble was requested. The authors indicated that was already being deployed and depicted in Figure 6. While I believe that I understand how this reduces biopsy times by the author's assertion, it also seems to be slightly confusing as presented in the Figure. Specifically, Figure 6a indicates 'stop when correctly detect any mutant'. This approach would confound the results and overemphasize the performance metrics. In reality you would not have the correctness or incorrectness to rely upon. Therefore, the biopsy would stop when any mutation was detected (right or wrong) and then you would determine the correctness of the model after the fact. If the stopping rules are dependent on the correctness, it seems it would exclude false positive examples increasing both the accuracy and the specificity.

This is indeed a limitation of the retrospective rechecking analysis. However, we believe it is currently the most appropriate approach to evaluate the effectiveness of CryoAID before initiating a real-world prospective study involving pathologists and patients. In the retrospective setting, the ground-truth mutation labels are available, allowing us to establish an idealized stopping rule to assess the *potential* of CryoAID in reducing re-biopsies.

In this framework, the “sampling” count can be viewed as the expected time to obtain a detection of the true mutation status, for both human expert and AI. When the model predicts an incorrect mutation status, it effectively fails to detect the *true* mutant and the sampling continues. This design quantifies an upper bound of CryoAID’s benefit under ideal conditions.

We fully acknowledge that the situation mentioned in your comment represents a more realistic clinical scenario. After demonstrating this potential under retrospective validation, we aim to

conduct prospective clinical trials in which pathologists collaborate with CryoAID without prior knowledge of the true labels. This setting is more complex but also clinically meaningful.

In the previous submitted version, we have already highlight the restriction of our retrospective rechecking analysis in Methods and Discussion.

“This statistic indicates the potential maximum reduction on re-biopsies when using CryoAID, while practical implementation relies on a proper collaboration between CryoAID and the pathologist.”

“Finally, we only investigated a retrospective scenario without the pathologist's assessment. Therefore, a prospective clinical trial implementing a pathologist-AI collaboration is expected to estimate CryoAID's real-world robustness in biopsy reductions.”

In this revised manuscript, we have clarified this limitation more explicitly in the Results section:

“Please note that this statistical process may be optimistic to a certain extent, as in real-world practice there is no ground-truth mutation status available during surgery, only the pathologist's judgment. We expect that the actual biopsy counts achieved through AI-pathologist collaboration will exceed the estimation in this analysis, but remain lower than those from human experts alone.”

Still quite a few copy editing errors:

Examples

1. Abstract:

Cryosection pathology is essential for intraoperative diagnosis of diffuse midline gliomas, yet it often leads to diagnostic errors and prompt unnecessary re-biopsies prior to the formal molecular assessment can be completed.

Cryosection pathology is essential for intraoperative diagnosis of diffuse midline gliomas, yet it often leads to diagnostic errors and could prompt unnecessary re-biopsies before the formal molecular assessment.

2. Flow chart in extended data '652 imaging highly suspected patients were remained' (and other mentions of 'were remained')

Revised to “remained” in all cases.

3. 'PAs 341 require maximal resection while an indication of H3K27M mutant could make surgeons to be conservative'

PAs require maximal resection, whereas the presence of an H3K27M mutation could make surgeons more conservative.

4. ..."CryoAID well detects the statuses in TP53"

CryoAID accurately detects the TP53, H3K27M, and ATRX statuses with AUC ...

There are quite a few additional grammatical errors that still make the manuscript difficult to follow.

We have revised above-mentioned issues in the paper and spent efforts to further clean up grammatical errors.

Reviewer #3 (Remarks to the Author):

1. Thank you for responding to the comments. Based on the authors' clarification, there does not appear to have been any misunderstanding in the summary section of my earlier remarks. The authors confirmed that the intended goal was in line with predicting "the mutation statuses of ATRX, H3K27M, and P53 using cryosection samples." This remark simply summarized the reported performance of such predictions in "Pass" images ("areas under the receiver operating characteristic curve (AUCs) ranging from 0.704 to 0.790") and "No Pass" images ("AUCs between 0.561 and 0.635") as presented in the original manuscript.
2. The discussion on "biopsy times" appears to have been copied and pasted from the response to Reviewer 2 and was not related to my comments. That said, this point was not central to understanding the authors' main message.
3. I am glad that the suggestions regarding foundation models contributed to substantial performance improvements and that the benchmarking analyses were useful in highlighting the strengths of the proposed approach.
4. The authors have adequately addressed the other previously raised comments. Thank you.

We sincerely appreciate your recognition and acceptance of our work.

Reviewer #3 (Remarks on code availability):

The code provides a README file with sufficient instructions for installing the application.